# Winter Wheat Yield Prediction Using an LSTM Model from MODIS LAI Products

**Jian Wang** [1], **Haiping Si** [1], **Zhao Gao** [2] **and Lei Shi** [1,*]

[1] College of Information and Management Science, Henan Agricultural University, Zhengzhou 450002, China
[2] The First Geodetic Survey Team of the Ministry of Natural Resources, Shaanxi Bureau of Surveying, Mapping and Geoinformation, Xi'an 710054, China
[*] Correspondence: shilei@henau.edu.cn; Tel.: +86-0371-56990030

**Abstract:** Yield estimation using remote sensing data is a research priority in modern agriculture. The rapid and accurate estimation of winter wheat yields over large areas is an important prerequisite for food security policy formulation and implementation. In most county-level yield estimation processes, multiple input data are used for yield prediction as much as possible, however, in some regions, data are more difficult to obtain, so we used the single-leaf area index (LAI) as input data for the model for yield prediction. In this study, the effects of different time steps as well as the LAI time series on the estimation results were analyzed for the properties of long short-term memory (LSTM), and multiple machine learning methods were compared with yield estimation models constructed by the LSTM networks. The results show that the accuracy of the yield estimation results using LSTM did not show an increasing trend with the increasing step size and data volume, while the yield estimation results of the LSTM were generally better than those of conventional machine learning methods, with the best $R^2$ and RMSE results of 0.87 and 522.3 kg/ha, respectively, in the comparison between predicted and actual yields. Although the use of LAI as a single input factor may cause yield uncertainty in some extreme years, it is a reliable and promising method for improving the yield estimation, which has important implications for crop yield forecasting, agricultural disaster monitoring, food trade policy, and food security early warning.

**Keywords:** winter wheat; yield estimation; LSTM; LAI; deep learning

## 1. Introduction

Wheat is an important crop in China, and its yield is directly related to the development of the national economy. Timely, accurate, and wide-ranging monitoring and forecasting of wheat yields is of great practical significance for national economic development and food policy formulation [1,2]. Due to its large coverage area and short detection period, satellite remote sensing provides a new technical tool for large-scale crop estimation and is rapidly becoming the most widely used technology in crop estimation.

At present, the methods of crop yield estimation using remote sensing technology can be broadly classified into three categories according to the characteristics of the models used: (1) the empirical modeling method; (2) the mechanistic modeling method; and (3) the semi-empirical (semi-mechanistic) modeling method. The empirical model directly uses spectral vegetation indices or canopy remote sensing inversion parameters to establish relationships with crop yields, which are characterized by their simplicity and ease, involving fewer crop yield formation mechanisms, and the relationships are generally established using conventional machine learning methods, such as support vector machine and random forest, with NDVI or leaf area index (LAI) as input parameters [3–8]. Such relationships are usually localized and difficult to generalize to other agricultural areas. Semi-empirical models and semi-mechanical models are also known as parametric models, among which the light energy utilization efficiency model is the most widely used [6,9–11], but some

parameters are difficult to quantitatively simulate. Mechanical models fully consider the mechanism of crop yield formation, but their solution process is complex and requires more input parameters, and some necessary parameters in the operation are difficult to obtain at a regional scale; meanwhile, most of these models estimate crop yields at the field scale [12–15]. Although the models work well for archiving yields, the accuracy may be reduced when scaled to the national level. Many a priori parameters are required in regional estimation. Due to the heterogeneity of the ground surface, the accuracy of the ground parameters is generally low, especially in the case of small farmland in China, resulting in low regional accuracy. Moreover, the computational process is complicated and requires many parameters, which can be limited in practical use.

In recent years, deep learning has been successfully applied to several fields, such as image recognition and language translation [11,16–22]. Compared with traditional machine learning methods, deep learning techniques often achieve better performance. CNN and recurrent neural network (RNN) are more widely used models in neural networks and have also been applied to crop yield estimation and prediction [19,23–29]. LSTM is a special kind of RNN [30,31], due to its recursive structure and gating mechanism that regulates the entry and exit of information into and out of cells, as well as its processing of sequential data. The LSTM has feedback connections and can handle the input sequences of arbitrary length and is often preferred in the classification, processing, and prediction based on time series data. Several studies used LSTM for crop yield prediction with impressive results. LSTM not only captures trends in the data but also describes the dependencies of the time series data. Tian et al. built an LSTM model by integrating the meteorological data and two remote sensing indices (vegetation temperature condition index (VTCI) and LAI) to estimate wheat yield in Guanzhong Plain [32]. Jeong et al. used water-related indices and the maximum temperature as inputs for rice yield prediction using an LSTM model, which showed reliable early prediction accuracy [16]. Sun et al. used a CNN-LSTM model to predict the end-of-season and in-season yields of soybean in the county. The input data for the model included meteorological data and MODIS surface temperature (LST) [24]. The LSTM model was shown to have high prediction accuracy for crop yield estimation, but all of the above methods for estimating crop yield use multiple data as input parameters, and it is difficult to obtain non-remote sensing data in some regions [33–35]. There are a large number of mature remote sensing products for LAI data, so this study mainly considered using LAI remote sensing products as single model input data and what kind of accuracy could be achieved when using deep learning algorithms for yield estimation.

In this study, LSTM was used to estimate the winter wheat yield at the county scale based on the relationship between time series LAI products and winter wheat yield. Considering the simplicity of obtaining LAI data, the model input parameters were only the leaf area index to verify the accuracy that could be achieved under the influence of a single factor. Moreover, the time step of the input remote sensing data of the model was considered so as to determine the accuracy in different time data.

## 2. Materials and Methods

### 2.1. Study Area

Henan Province is located in eastern central China (31°23′~36°22′ N, 110°21′~116°39′ E). Figure 1 is a schematic diagram of the location of the study area. Henan Province is located between the warm temperate zone and the subtropical zone. The terrain of Henan Province is high in the west and low in the east, with mountains above 1000 m above sea level in the west and plains below 100 m in the east. Mountains and hills account for 44.3% of the total area, and plains account for 55.7%. The total wheat output of Henan Province ranks first in China, accounting for more than 28% of the country's total wheat output. The sowing time of wheat varies from north to south by nearly two weeks, and there is a large gap in yield in different regions. The topography of Henan Province is complex and diverse, the topography is low in the east and high in the west, with significant differences; the surface morphology is complex and diverse. Due to the influence of landforms and

monsoons, Henan Province has a wide variety of soil types and large differences in climate resources, resulting in significant differences in crop yields in different regions. According to the production conditions of producing areas and the climatic characteristics of the wheat growth period, the wheat-growing areas in Henan Province can be divided into five regions.

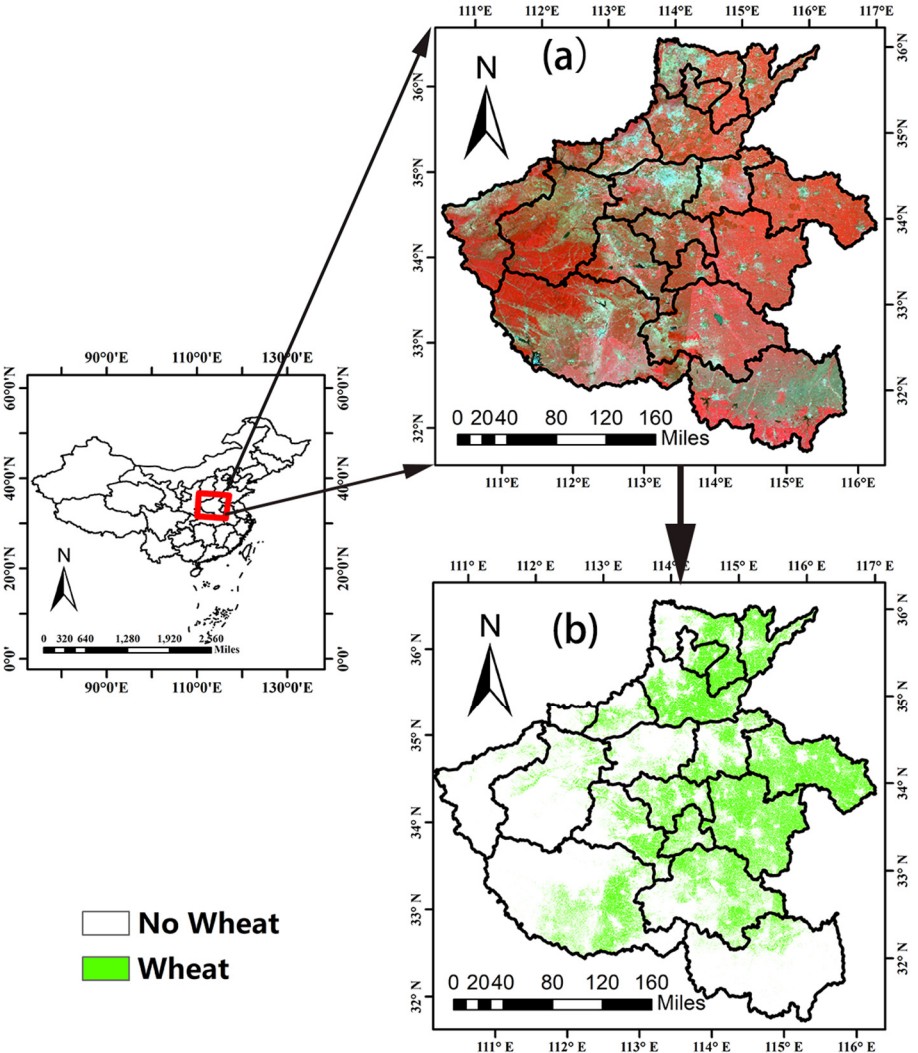

**Figure 1.** The geographic location of the study area and classification results (The red box on the left is the location of Henan Province. (**a**)—false color composite with the MODIS09 data; and (**b**)—land classification, green represents the winter wheat region).

1.  The wheat area in Nanyang Basin, including Nanyang City and Biyang Countyin Zhumadian, is a typical rainfed and semi-rainfed area due to the relatively poor field supporting projects and irrigation conditions.
2.  Rice stubble and wheat areas in southern Henan, including Xinyang, southern Zhumadian, and Nanyang Tongbai. The soil in this area is heavy, and the precipitation during the wheat growth period is relatively high.
3.  In western Henan, southwestern Henan, and northern Henan, dry wheat areas, including Luoyang, Sanmenxia, Jiyuan, Pingdingshan, Anyang, and other shallow hilly areas, drought, winter, and spring freezing damage, rust, and yellow dwarf disease are the main factors affecting wheat yield.
4.  The wheat area in north-central Henan Province, including Xuchang, Zhengzhou, Luoyang, and the irrigated land north of the Yellow River, has good production conditions and high production levels.

5. The wheat area in the central and eastern part of Henan Province includes the irrigated land in the middle- and high-yield wheat areas in the north-central part of Zhumadian, Luohe, Zhoukou, Shangqiu, and Pingdingshan Mountain.

The topography and climate within each of the above production areas are relatively consistent; so, we developed a winter wheat yield model for each production area. The actual yield data for winter wheat in this study were provided by the Henan Provincial Bureau of Statistics.

### 2.2. MODIS LAI

LAI is defined as the area of unilateral green leaves per unit of ground area in a broadleaf canopy and half of the total needle surface area per unit ground area in a coniferous canopy. The LAI product selected for this study was MCD15A2H. MCD15A2H is an 8-day composite product with a total of 46 scenes per year and a spatial resolution of 500 m. The inversion algorithm for the Moderate Resolution Imaging Spectroradiometer (MODIS) LAI product was a look-up table constructed based on a three-dimensional radiative transfer model. When the main algorithm failed, a backup algorithm using an empirical relationship between NDVI and the canopy LAI was triggered to estimate the LAI for each pixel, and the look-up table was used to compare whether the observed and simulated canopy top BRFs were within a given biologically relevant range. All canopy/soil patterns and corresponding LAI values that differed between the modeled and observed BRFs within a given level of uncertainty were considered acceptable solutions. The product used for this study was version 6, and the final result was the true LAI.

All satellite data are archived in HDF-EOS format, and the MODIS Reprojection Tool (MRT) software provided by NASA enables the user to read the HDF-EOS format. This software supports the performing geo-transformations to different coordinate systems or cartographic projections and writing the output to other file formats (GeoTIFF). All data are initially projected onto an integer sine wave (ISIN) mapping grid. These data were corrected to UTM coordinates using MRT and resampled using the nearest neighbor algorithm. Non-wheat fields were masked using a land cover classification, and then the corresponding areas were cut out of the remotely sensed imagery using SHP data based on the extent of each city. The MODIS LAI products include the quality control (QC) information designed to help users make the best use of these data. Each QC layer has a large amount of quality information associated with each pixel, whether the pixel is labeled as cloudy, clear, or in cloud shadow. In a subsequent study, MODIS LAI selected high-quality pixels derived using the master algorithm under cloud-free conditions. Figure 2 shows the time series results of the MODIS LAI for the entire winter wheat growing region in Henan, and the maximum value of the LAI was approximately 2. Considering that LAI is at a low level in the early and late stages of wheat growth, and that these growth stages do not have a particularly strong influence on yield formation, we used LAI from flowering to maturity each year as input data.

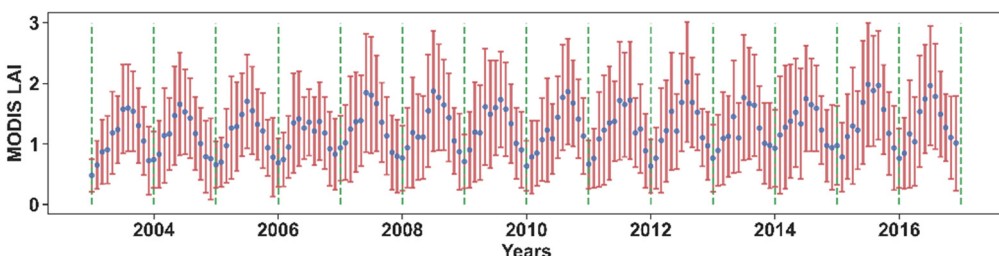

**Figure 2.** The mean curve of MODIS LAI in Henan Province from 2003 to 2016. The red line is the error bars plotted using standard deviation; blue dots represent the mean; between the two green dotted lines is the LAI for one year.

### 2.3. Mapping of Wheat Distribution

This study focused on winter wheat, which first required the extraction of winter wheat distribution areas from remote sensing images. We used the product produced by Dong [36]. The product was produced by first synthesizing monthly NDVI maxima and using FROM-GLC as a priori knowledge to obtain crop distribution information; then, we used the standard seasonal growth curve of winter wheat combined with the time-weighted dynamic time warping (TWDTW) to determine the area of winter wheat. The accuracy of the final product was higher than 89.30% and 90.59% for producers and users, respectively. To maintain consistency with the resolution of MODIS data, the classification results were finally resampled to 500 m using the mode resampling method. Mode resampling selects the value with the highest frequency of occurrence among all sampling points, and the results maintain the real state of the ground surface to some extent. The classification results are shown in Figure 1b.

### 2.4. LSTM

The recurrent neural network (RNN) is a type of neural network with short-term memory capabilities. In a cyclic neural network, a neuron cannot only receive information from other neurons but also its own information, forming a network structure with loops. Compared with feedforward neural networks, recurrent neural networks are more in line with the structure of biological neural networks. Recurrent neural networks have been widely used in tasks such as speech recognition, language modeling, and natural language generation. The parameter learning of recurrent neural networks can be learned by the backpropagation algorithm over time. The backpropagation algorithm with time transmits the error information step by step in the reverse order of time. When the input sequence is relatively long, there will be the problem of gradient explosion and disappearance. In order to solve this problem, people have made many improvements to the cyclic neural network. The most effective means of improvement is to introduce a gating mechanism, one of which is called a long short-term memory network (LSTM). LSTM is a variant of a cyclic neural network, which can effectively solve the problem of gradient explosion or the disappearance of a simple cyclic neural network.

The ingenuity of LSTM is that, by increasing the input threshold, the forgetting threshold, and output threshold, the weight of the self-loop is changed. As such, when the model parameters are fixed, the integration scale at different times can be dynamically changed, thereby avoiding the problem of gradient disappearance or gradient expansion.

The LSTM network introduces a gating mechanism to control the path of information transmission. The three "gates" are the input gate $i_t$, the forget gate $f_t$, and the output gate $o_t$. The functions of these three gates are as follows:

(1) The forgetting gate $f_t$ controls how much information needs to be forgotten to control the internal state $c_{t-1}$ at the last moment;

(2) The enter gate $i_t$ controls the candidate state $\widetilde{c}_t$ at the current moment and how much information needs to be saved;

(3) The output gate $o_t$ controls how much information of the internal state $c_t$ at the current moment needs to be output to the external state $h_t$.

When $f_t = 0$ and $i_t = 1$, the memory unit clears the historical information and writes the candidate state vector $\widetilde{c}_t$. However, the memory unit $c_t$ is still related to the historical information at the previous moment. When $f_t = 1$ and $i_t = 0$, the memory unit will copy the content of the previous moment without writing new information.

Figure 3 shows the cyclic unit structure of the LSTM network. The calculation process is: (1) first use the external state $h_{t-1}$ at the previous moment and the input $x_t$ at the current moment to calculate the three gates and the candidate state $\widetilde{c}_t$; (2) combine the forget gate $f_t$ and the input gate $i_t$ to update the memory unit $c_t$; and (3) combine the output gate $o_t$ to transfer the information of the internal state to the external state $h_t$.

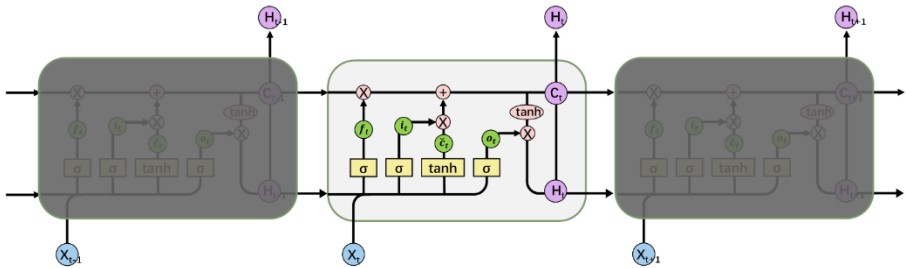

**Figure 3.** Cyclic cell structure of the LSTM network.

By means of LSTM cyclic units, the whole network can be built up with long-distance temporal dependencies. It can be succinctly described as

$$
\begin{bmatrix} \widetilde{c}_t \\ o_t \\ i_t \\ f_t \end{bmatrix} = \begin{bmatrix} tanh \\ \sigma \\ \sigma \\ \sigma \end{bmatrix} \left( w \begin{bmatrix} x_t \\ h_{t-1} \end{bmatrix} + b \right) \tag{1}
$$

$$
c_t = f_t \odot c_{t-1} + i_t \odot \widetilde{c}_t \tag{2}
$$

$$
h_t = o_t \odot \tan \mathrm{h}(c_t) \tag{3}
$$

In the LSTM network, a memory unit c can capture a key piece of information at a certain moment and can store this key information for a certain time interval. The lifetime of information stored in memory unit c is longer than that of short-term memory h but much shorter than that of long-term memory. Figure 4 shows the workflow from LAI data processing to LSTM model building and yield estimation.

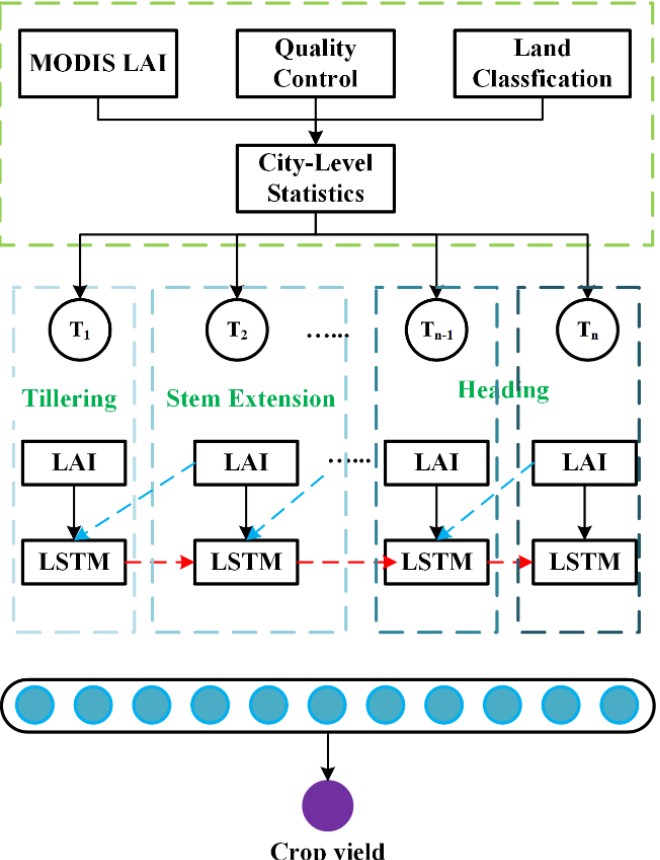

**Figure 4.** Overall structure of the LSTM model for wheat yield estimation.

*2.5. Model Evaluation Metrics*

We chose root mean square error (RMSE), coefficient of determination ($R^2$), and mean absolute percentage error (MAPE), the three metrics in Equations (4)–(6). To evaluate the performance of the model, the difference between the predicted and actual statistical values of the model was calculated. A smaller RMSE indicates a better performance of the model, and a larger $R^2$ indicates a higher regression accuracy of the model. The lower the value of MAPE and RMSE, the higher the accuracy of the obtained predictive model. MAPE measures the error in percentage and specifies the average percentage deviation between the forecast value and the actual implementation [37]. Usually, the fit of the model is perfect when the MAPE value is below 10% and when it is in the range from 10% to 20%, the model fit is good. In the range of 20–30%, the error level is acceptable and when it exceeds 30%, the model is a poor fit and should be rejected [38].

$$\text{RMSE} = \sqrt{\frac{1}{n} \sum_{i=1}^{n} (\hat{y}_i - y_i)^2} \tag{4}$$

$$R^2 = 1 - \frac{\sum_{i=1}^{n} (\hat{y}_i - y_i)^2}{\sum_{i=1}^{n} (y_i - \hat{y}_i)^2}, \tag{5}$$

$$\text{MAPE} = \frac{100\%}{n} \sum_{i=1}^{n} \left| \frac{\hat{y}_i - y_i}{y_i} \right|, \tag{6}$$

where $y_i$ represents the actual yield, $\hat{y}_i$ is the predicted yield, and $n$ is the sample size.

## 3. Results and Discussion

*3.1. Yield Estimation under Different Time Steps and Input Combinations*

We used winter wheat LAI time series data from 2003 to 2015 as training samples and modeled them with LSTM to obtain the winter wheat yield prediction results for 2016 and compared the predicted yield with the statistical yield in 2016 to verify the prediction accuracy of the model. Moreover, considering the characteristics of the LSTM model and the requirements of the yield prediction task, the input data required for training were processed as subsequently described, and the winter wheat growth data from March to the end of May each year were selected so that the input data for one year had 12 LAI values containing data from the flowering stage to the maturity stage. The fitted data of the model changed at different input steps, which also had an impact on the accuracy of the prediction results. To obtain the input step with the highest accuracy, the input steps from 1 to 6 were compared. The overall RMSE for Henan Province under the six step-size scenarios is shown in Figure 5. The solid red line in the figure is the 1:1 line, and the estimation results were mostly evenly distributed on both sides with the solid line as the center, indicating that the model had a good prediction for the yield. However, when the yield exceeded 7000 kg/ha, the estimation accuracy of the model tended to decrease, and the yield prediction tended to be underestimated, which is consistent with the systematic trend of underestimation at high yields in previous studies, mainly due to the relatively low proportion of data from high production areas in the sample. We first constructed LSTM models for five winter wheat production areas and finally obtained the winter wheat yield estimation results for the whole of Henan Province. A–E in Table 1 show the RMSE, $R^2$ and MAPE of the five grain-producing regions at different step sizes, and F shows the statistical results for the whole Henan Province. Figure 6 shows the RMSE histograms of the LSTM model for the five regions and the whole of Henan Province under different step lengths. When the step size was short, the predicted yields of the LSTM model for different production areas had great instability compared with the statistical yields in the field, but when the step size increased to 3 and 4, the accuracy started to improve and stabilized. However, the accuracy decreased again when the step size was 6. The LSTM model achieved optimal values of $R^2$ and RMSE at a step size of 4, with results of 88% and 532.16 kg/ha, respectively, while the value of MAPE was 4.44%, which was at a very high level of fit. Although the problem of

LSTM was effectively solved in terms of gradient disappearance or explosion compared with the general RNN networks, the winter wheat yields were not closely related to yields from many years ago, and to some extent, were only strongly correlated with historical yields from the last three or four years, and there may be a loss of accuracy if the current results are fitted with data from a longer period of time ago.

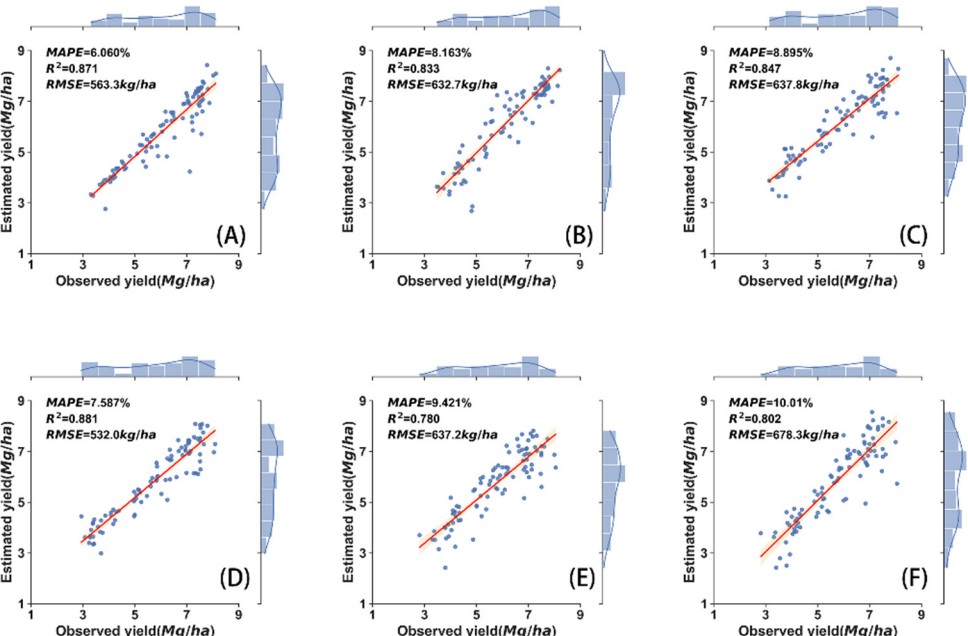

**Figure 5.** Comparison of yield estimation results for LSTM models based on different time steps: (**A**) one time step; (**B**) two time steps; (**C**) three time steps; (**D**) four time steps; (**E**) five time steps; and (**F**) six time steps. The probability distributions of the statistical and estimated yields are plotted on the right and top of the Y axis, respectively.

**Table 1.** The accuracy evaluation of the LSTM prediction model was compared between different time steps when 12 LAI data were input.

| Time Steps | RMSE (kg/ha) | | | | | | $R^2$ | | | | | | MAPE (%) | | | | |
|---|---|---|---|---|---|---|---|---|---|---|---|---|---|---|---|---|---|
| | A | B | C | D | E | F | A | B | C | D | E | F | A | B | C | D | E |
| 1 | 1039.93 | 436.63 | 350.59 | 386.97 | 687.70 | 563.38 | 0.40 | 0.79 | 0.95 | 0.92 | 0.01 | 0.87 | 12.33 | 5.01 | 4.28 | 5.19 | 7.29 |
| 2 | 468.29 | 548.72 | 831.34 | 699.47 | 219.07 | 632.77 | 0.86 | 0.73 | 0.74 | 0.81 | 0.27 | 0.83 | 7.82 | 7.72 | 10.57 | 12.40 | 2.31 |
| 3 | 604.58 | 589.19 | 635.35 | 767.26 | 583.36 | 637.88 | 0.86 | 0.78 | 0.87 | 0.82 | 0.03 | 0.85 | 10.17 | 9.29 | 8.79 | 12.63 | 5.90 |
| 4 | 530.79 | 461.56 | 568.10 | 667.68 | 396.56 | 532.08 | 0.83 | 0.78 | 0.87 | 0.90 | 0.38 | 0.88 | 7.56 | 7.24 | 8.02 | 11.75 | 4.44 |
| 5 | 393.49 | 549.11 | 635.34 | 699.27 | 713.08 | 637.27 | 0.90 | 0.74 | 0.80 | 0.70 | 0.00 | 0.78 | 6.99 | 8.70 | 9.40 | 13.01 | 8.45 |
| 6 | 795.51 | 414.61 | 770.62 | 548.72 | 674.49 | 678.36 | 0.86 | 0.80 | 0.78 | 0.81 | 0.24 | 0.80 | 13.41 | 6.96 | 12.04 | 9.17 | 7.95 |

The respective highest accuracy estimation results in different regions differed significantly in time steps. For the whole of Henan Province, the overall accuracy may not be optimal if the same step was used for yield estimation. The lowest value of RMSE can be seen in Figure 6 in the southwestern production area, with an RMSE of approximately 400, and the highest in time steps was in southwestern Henan, with approximately 700. There was a correlation between the winter wheat yield and time series LAI data, but the interannual correlation was to some extent not a more accurate result by modeling with more data, and there are many factors affecting the wheat yield, including variation in variety, temperature, and topography, which can all cause yield fluctuations, although all these factors can affect the LAI time series to some extent. However, when using time series LAI data alone as input parameters, it is not better to use more data, and it is not better to set a higher time step; these should be quantitatively and individually tested for different regions and not generalized.

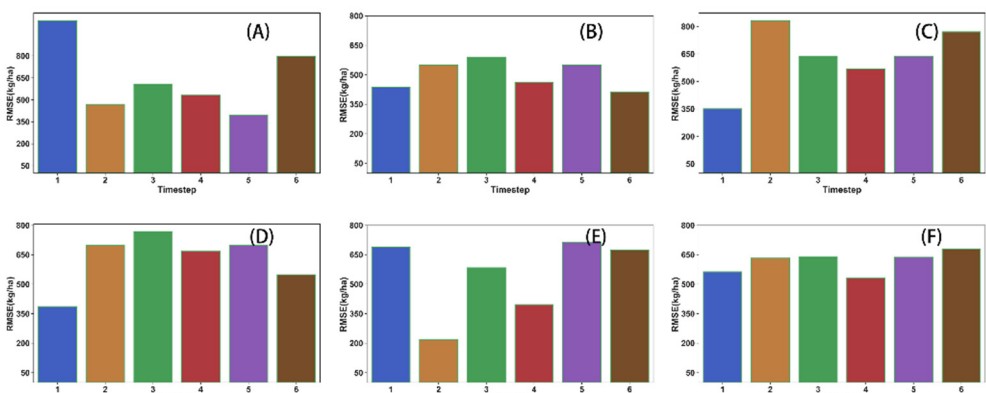

**Figure 6.** The model performance (predicted RMSE) using different time steps for the whole growing season. (**A**) one time step; (**B**) two time steps; (**C**) three time steps; (**D**) four time steps; (**E**) five time steps; and (**F**) six time steps.

### 3.2. Yield Estimation with Different Input Time Series Data

In the process of winter wheat yield estimation, it is always desirable to predict the yield as early as possible. Therefore, we considered shortening the time range of the LAI data used and the used data from the plucking stage to the filling stage for modeling so that their yield could be predicted 16 days before maturity. The LAI data used in this study were only six per year, which halved the amount of data compared to previous studies, making it easier to obtain the data, especially the high spatial resolution satellite data. The results of the comparison between the predicted yield and the actual statistical yield are shown in Figure 7, from which there was no significant decrease in the prediction accuracy for all of Henan compared to the 12 data, and the accuracy of individual time steps increased. The best time step for the overall accuracy occurred at 2, where the $R^2$ and RMSE were 87% and 522.32 kg/ha, respectively, while MAPE was 5.67%. A–E in Table 2 shows the RMSE, $R^2$ and MAPE of the five grain-producing regions at different step sizes, and F shows the statistical results for the whole Henan Province; Figure 8 shows the histogram of RMSE. However, it can also be seen that the value of RMSE gradually increased with increasing step size. This may be due to the fact that LAI data were strongly correlated only in the last two years, and the larger the time difference, the lower the correlation between LAI and yield; moreover, the interannual LAI was not strongly correlated with early and late yield formation, so in the process of yield estimation using time series LAI, a higher accuracy was obtained by using data from the nodulation to filling stage. It has been shown that the accumulation of dry matter in winter wheat is mainly concentrated at the nodulation and gestation stages, and the LAI in this period was closely correlated with the yield formation of winter wheat, which is also more consistent with the results of this study. It should also be noted that the overall RMSE accuracy was the best except for the case of step size 2. The RMSE of the remaining steps was significantly lower compared to the model with 12 data; therefore, using as many growing period data as possible would also make the model more robust when using time-series LAI for yield estimation.

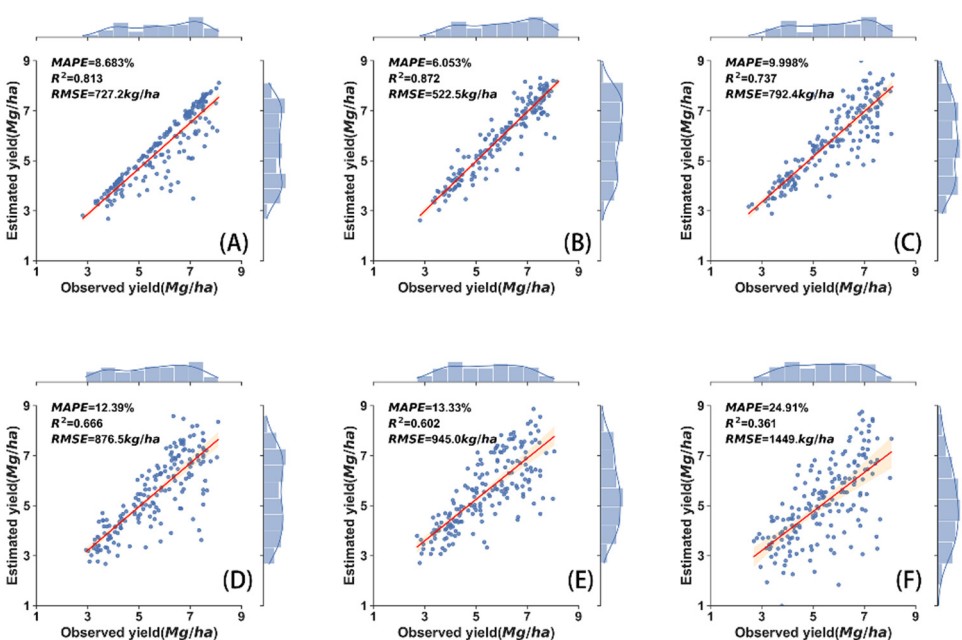

**Figure 7.** Comparison of the yield estimation results for LSTM models based on different time steps (**A**) one time step; (**B**) two time steps; (**C**) three time steps; (**D**) four time steps; (**E**) five time steps; and (**F**) six time steps. The probability distributions of the statistical and estimated yields are plotted on the right and top of the Y axis, respectively.

**Table 2.** The accuracy evaluation of the LSTM prediction model was compared between different time steps when 6 LAI data were input.

| Time Steps | RMSE (kg/ha) | | | | | | $R^2$ | | | | | | MAPE (%) | | | | |
|---|---|---|---|---|---|---|---|---|---|---|---|---|---|---|---|---|---|
| | A | B | C | D | E | F | A | B | C | D | E | F | A | B | C | D | E |
| 1 | 1180.61 | 534.22 | 798.75 | 338.77 | 656.31 | 727.11 | 0.17 | 0.70 | 0.80 | 0.96 | 0.19 | 0.81 | 17.63 | 5.22 | 10.29 | 5.67 | 6.64 |
| 2 | 372.66 | 552.27 | 534.55 | 411.21 | 598.78 | 522.32 | 0.87 | 0.67 | 0.86 | 0.93 | 0.18 | 0.87 | 6.03 | 7.41 | 5.61 | 6.53 | 5.67 |
| 3 | 704.62 | 362.19 | 867.82 | 739.41 | 901.01 | 792.37 | 0.53 | 0.85 | 0.75 | 0.67 | 0.01 | 0.74 | 9.78 | 4.63 | 11.11 | 12.23 | 9.83 |
| 4 | 1010.14 | 789.28 | 800.46 | 850.57 | 978.56 | 876.48 | 0.18 | 0.31 | 0.74 | 0.63 | 0.00 | 0.67 | 14.36 | 12.82 | 11.30 | 12.37 | 12.99 |
| 5 | 743.15 | 591.05 | 928.28 | 743.79 | 1255.65 | 945.34 | 0.44 | 0.61 | 0.63 | 0.67 | 0.00 | 0.60 | 11.69 | 10.62 | 12.73 | 12.17 | 17.03 |
| 6 | 1052.22 | 1572.15 | 1170.50 | 1167.80 | 1954.85 | 1448.93 | 0.28 | 0.02 | 0.50 | 0.36 | 0.02 | 0.36 | 20.81 | 45.16 | 17.92 | 21.91 | 27.96 |

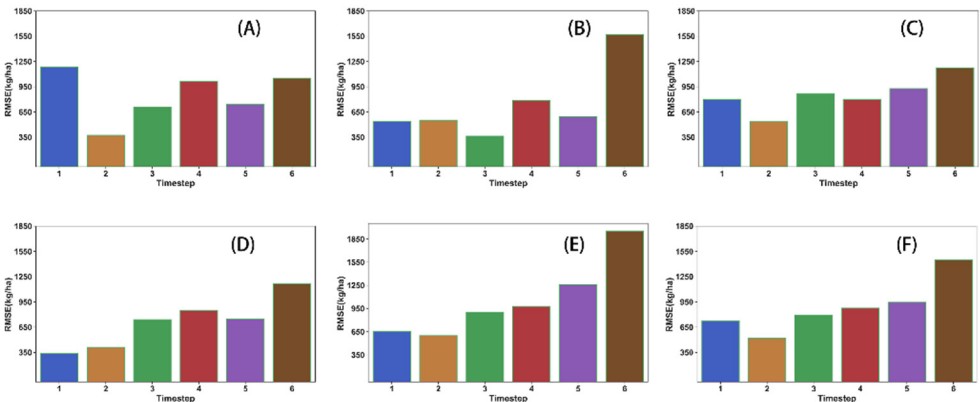

**Figure 8.** The model performance (predicted RMSE) using different time steps for only one period of the growing season. (**A**) one time step; (**B**) two time steps; (**C**) three time steps; (**D**) four time steps; (**E**) five time steps; and (**F**) six time steps.

### 3.3. Performance Comparison with Machine Learning Methods

Here, four machine learning methods (random forest, support vector regression, partial least squares regression, and XGBoost) were used to construct county-level wheat yield

models for each agroecological zone. Random forest (RF) is a supervised machine learning algorithm based on integration learning [39]. Different subsets are randomly drawn from the provided data and used to build several different decision trees and integrate the results of one decision tree according to Bagging's rules. Support vector regression (SVR) is a regression algorithm that is a variant of SVM in regression analysis [40]. SVR also considers maximization intervals but considers points within the decision boundary so that as many sample points as possible lie within the interval. The partial least squares regression (PLSR) [41] algorithm is a regression modeling method for multiple dependent variables Y on multiple independent variables X. The algorithm considers extracting as many principal components as possible from Y and X in building the regression and maximizing the correlation between the principal components extracted from X and Y, respectively. XGBoost [42] is a scalable machine learning system that adds to the objective function of each iteration regular term to further reduce the risk of overfitting, XGBoost is an all-in-one machine learning algorithm.

The yield prediction of winter wheat in Henan Province was constructed using the four methods mentioned above. The prediction model first used winter wheat LAI time series data from 2003 to 2015, followed by yield prediction for 2016, and the $R^2$ and RMSE of the prediction results are shown in Table 3. For the whole of Henan Province, the best performance among the four methods was the SVR with R2, RMSE and MAPE of 0.76, 725.8 kg/ha and 6.33%, respectively, and the worst was PLSR with R2, RMSE and MAPE of 0.7 and 809.1 kg/ha and 7.74%, respectively. Compared with these machine learning methods, the prediction results of the LSTM had better accuracy and performance both for individual wheat growing areas and for the whole of Henan province.

**Table 3.** Accuracy evaluations comparison among different methods.

| Model | RMSE (kg/ha) | | | | | | $R^2$ | | | | | | MAPE (%) | | | | |
|---|---|---|---|---|---|---|---|---|---|---|---|---|---|---|---|---|---|
| | A | B | C | D | E | F | A | B | C | D | E | F | A | B | C | D | E |
| RF | 566.2 | 705.8 | 946.5 | 693.3 | 656.6 | 774.5 | 0.66 | 0.37 | 0.62 | 0.73 | 0.35 | 0.72 | 9.31 | 9.48 | 9.05 | 11.24 | 7.04 |
| SVR | 605.0 | 558.9 | 980.0 | 545.2 | 505.4 | 725.8 | 0.60 | 0.56 | 0.60 | 0.83 | 0.61 | 0.76 | 10.26 | 8.53 | 10.26 | 9.31 | 6.33 |
| PLSR | 627.7 | 680.4 | 1051.4 | 558.2 | 676.8 | 809.1 | 0.57 | 0.37 | 0.55 | 0.82 | 0.31 | 0.70 | 11.03 | 11.18 | 11.28 | 9.06 | 7.74 |
| XGBOOST | 579.2 | 659.4 | 976.4 | 737.8 | 638.1 | 785.8 | 0.62 | 0.45 | 0.60 | 0.69 | 0.38 | 0.72 | 8.62 | 10.40 | 10.12 | 11.60 | 6.98 |

The prediction accuracy of the four machine learning methods was the lowest in the southwest Henan region, which is a mountainous and hilly area with complex topography, a small winter wheat growing area, and large yield variation, and there may be a lack of yield accuracy in this region using conventional methods to construct the model. Compared with these, using LSTM model for winter wheat yield prediction had a superior performance. Compared to algorithms such as SVR and RF, the accuracy of estimation was not sufficient, because the ability to analyze complex nonlinear relationships between long time series variables is not as good as LSTM, resulting in poor model performance. These machine learning methods do not consider the time correlation between winter wheat yields in the modeling process, and the estimation for each year's yield is conducted independently, while LSTM takes into account the time series correlation of yields and also can better handle the nonlinear relationship, so it has higher accuracy compared to machine learning. Overall, a better performance capability can be obtained using LSTM models for forecasting time series data.

## 4. Conclusions

In this study, considering the complexity of data collection, we used LAI as a single input variable and five models, including four machine learning models (RF, SVR, PLSR, and XGBOOST) and one deep learning model (LSTM) to predict the winter wheat yield in Henan Province in 2016. In general, the LSTM model had superior performance compared with the machine learning models. Moreover, considering the characteristics of the LSTM, the time step of the modeled data as well as the growth period data were analyzed, and the

time step needs to be analyzed for different growth regions, while using only the necessary growth period data can also obtain a high prediction accuracy; however, for the robustness of the model, the more growth period data are used accordingly, the better. To date, winter wheat yield prediction based on remote sensing images has been carried out at the county level. However, the determination of crop yield remains a challenge because the variability and uncertainty within the region is unknown. The results of our study on winter wheat yield prediction at a regional scale using publicly available data, using LAI as an input variable for determining crop yield, can potentially be applied to crop yield estimation in regions with sparse observational data and worldwide.

**Author Contributions:** Conceptualization, J.W.; resources, Z.G.; writing—original draft preparation, J.W.; writing—review and editing, J.W., L.S. and H.S. All authors have read and agreed to the published version of the manuscript.

**Funding:** This research was funded by the National Natural Science Foundation of China (NO.42101362, 31501225); the Natural Science Foundation of Henan Province of China (NO.222300420463).

**Institutional Review Board Statement:** Not applicable.

**Data Availability Statement:** Not applicable.

**Conflicts of Interest:** The authors declare no conflict of interest.

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
