# Peer review of "Winter Wheat Yield Prediction Using an LSTM Model from MODIS LAI Products"

_agriculture, doi:10.3390/agriculture12101707_

Round 1

Reviewer 1 Report

Dear Authors,

I revised the manuscript "Winter Wheat Yield Prediction using an LSTM Model from MODIS LAI Products" submitted to the Agriculture journal. The manuscript is interesting. Unfortunately, the weakest part of the manuscript is the discussion of results. In addition, I have some concerns which need to be addressed before considering for final publication.

Minor comments:

1. Section "2. Materials and Methods":

a) In this section, also add information about the research's date. Currently, it can only be found on line 232.

b) In addition, the description and equations of the RMSE and R2 statistical indicators are missing. It is worth adding another prediction error indicator, MAPE because it is expressed as a percentage. For more information on this indicator, see the paper https://doi.org/10.3390/land10060609.

2. Line 210. The correct figure number is 3.

3. Line 222-224. This part of the text about LSTM networks is not necessary here.

Major comment:

Section "3. Results and Discussion". The discussion of the results should be greatly expanded. You need to compare your results to the results from other papers. Scientific work needs a real discussion of results! Additional references are also needed.

Author Response

Thank you very much for your comments and suggestions on our paper. They were very useful in revising the manuscript. Our item-by-item response to your comments is as follows, where our response is in red fonts. The modifications of the paper are marked by a yellow background.

Reviewer 2 Report

The paper is a solid piece of work on applying a deep learning approach to the question of yield forecast. What makes the paper unique enough to be worth publishing is the applied LSTM approach to deep learning. Although it has been successfully applied to a large palette of problem, yield forecast was not yet among them.

The paper is well written, the content is clearly presented. Before publishing though, the applied LSTM-approach and its specific suitability for the given problem of yield forecast should be explained in more depth in the paper.

Also please also discuss shortly the practical implications of the fact, that  the approach seems to be very well suited for "normal" years and looses accuracy for exceptional years. Does that mean that it is not well suited for forecasts since forecasting the yield of a a normal year means forecasting the alreay expected?    

Formally, please clarify the y-axis in Fig2. The values of LAI are clearly out of range. Also, the axis caption size is much too small to be readable. 

Author Response

(The authors gave the same response as above.)
